# Review of Non-Destructive Civil Infrastructure Evaluation for Bridges: State-of-the-Art Robotic Platforms, Sensors and Algorithms

**DOI:** 10.3390/s20143954

**Published:** 2020-07-16

**Authors:** Habib Ahmed, Hung Manh La, Nenad Gucunski

**Affiliations:** 1Advanced Robotics and Automation Lab, Department of Computer Science and Engineering, University of Nevada, Reno, NV 89512, USA; hahmed@nevada.unr.edu; 2Department of Civil and Environmental Engineering, Rutgers, the State of University of New Jersey, Piscataway, NJ 08854, USA; gucunski@soe.rutgers.edu

**Keywords:** non-destructive evaluation (NDE), structural health monitoring (SHM), electric resistivity (ER) sensors, ground-penetrating radar (GPR), infrared (IR) thermography, impact echo (IE), NDE sensor fusion, convolutional neural network (CNNs), concrete crack detection, rebar detection and localization

## Abstract

The non-destructive evaluation (NDE) of civil infrastructure has been an active area of research in recent decades. The traditional inspection of civil infrastructure mostly relies on visual inspection using human inspectors. To facilitate this process, different sensors for data collection and techniques for data analyses have been used to effectively carry out this task in an automated fashion. This review-based study will examine some of the recent developments in the field of autonomous robotic platforms for NDE and the structural health monitoring (SHM) of bridges. Some of the salient features of this review-based study will be discussed in the light of the existing surveys and reviews that have been published in the recent past, which will enable the clarification regarding the novelty of the present review-based study. The review methodology will be discussed in sufficient depth, which will provide insights regarding some of the primary aspects of the review methodology followed by this review-based study. In order to provide an in-depth examination of the state-of-the-art, the current research will examine the three major research streams. The first stream relates to technological robotic platforms developed for NDE of bridges. The second stream of literature examines myriad sensors used for the development of robotic platforms for the NDE of bridges. The third stream of literature highlights different algorithms for the surface- and sub-surface-level analysis of bridges that have been developed by studies in the past. A number of challenges towards the development of robotic platforms have also been discussed.

## 1. Introduction

The monitoring, maintenance and rehabilitation of civil infrastructure is of paramount importance at the national and international level. Of the different types of civil infrastructure, the need for the maintenance and evaluation of bridges has been stressed by studies in the recent past [1,2,3,4]. The need for public infrastructure evaluation and monitoring is important, as a large number of civilian populations use the different infrastructures on a daily basis. Any structural defects that remain unchecked for a long time can lead to serious hazards for civilians utilizing that particular infrastructure. For the case of bridges alone, according to the National Bridge Inventory (NBI) statistics, there are more than 607,380 bridges in the entirety of the United States [5]. Although the overall ratio of marginally or seriously damaged bridges has been declining in recent decades, the recent statistics outlined by the U.S. Department of Transportation have classified around 67,000 bridges as structurally deficient and around 85,000 as functionally obsolete in nature [5]. Out of the $14.3 billion expenditure sanctioned for the maintenance of existing bridges and the construction of new bridges in 2010, $12.8 billion was dedicated towards the maintenance of existing bridges [6], which shows that a considerable portion of annually allocated funds are being diverted for the maintenance of bridges. A number of different factors contribute towards the partial or total destruction of bridges, ranging from design errors and construction defects to environmental degradation, scour, flood, collision and overloading [7,8]. The impact of bridge destruction and collapse far exceeds the overall material and financial costs associated with the bridge construction, as it also includes the various direct and indirect costs, which include, but are not limited to, the loss of lives, user delays, planning for alternate routes, along with the green-house gas emissions linked to detours and delays in traffic [7,9,10,11]. Figure 1 highlights the multitude of bridge destruction incidents in the U.S. in recent decades. Figure 2 provides a flow chart with an examination of the inter-relationship between the different sections of the review-based study. Table 1 outlines a complete list of abbreviations used in this review paper. It is being predicted that with the increase in climate change and frequency of adverse climate incidents (e.g., hurricane, floods, tsunamis) on a global scale, the overall costs related to bridge repair and maintenance is also expected to accelerate from $140 billion to $250 billion annually [10] with direct and indirect losses amounting to more than 17% of the total expenditures [11]. Therefore, the timely evaluation, monitoring and rehabilitation of bridges can result in reduced overall direct costs as well as the indirect costs in terms of the potential destruction of property and lives incurred in the wake of bridge destruction. Although natural disasters cannot be averted, but the different techniques for non-destructive evaluation (NDE) have the potential towards minimizing the overall direct and indirect costs associated with the destruction of bridges caused by internal deficiencies, construction deficits and maintenance-related issues. In the light of this realization, a number of national-level initiatives have been developed in the United States. One such example is the Long-Term Bridge Performance Program (LTBP) initiated by the Federal Highway Administration (FHWA) with the primary aim towards promoting the utilization of non-destructive evaluation technologies and techniques for regular bridge inspection and maintenance [12].

Apart from the increased costs associated with the destruction of bridges and other civil infrastructures, there are a number of different factors serving as valid motivations to perform a comprehensive review of different state-of-the-art solutions for the NDE of bridges. In the past decade, there has been an increase in the development of various automated robotic platforms with different capabilities and functionalities. Therefore, it is important to examine the various aspects of existing robotic platforms, which can serve as a benchmark for guiding research in the NDE of infrastructure in the future. Many of the relevant review-based studies in the recent past that focus on the robotic platforms in the field of NDE fail to provide a holistic evaluation of the different critical aspects, namely the type of robotic platforms used, the sensory modalities employed for NDE data collection and the data analysis techniques leveraged for accurately analyzing the collected data. These review-based studies only emphasize a limited number of studies. In this regard, this review-based study will provide insights on a wide-range of novel robotic platforms (e.g., ground robots, aerial robots and marine robots), along with the different sensory modalities (e.g., visual, acoustic, electric and contact-based sensors) and data analysis techniques for examining the surface-level and sub-surface-level defects that have been employed for analyzing the structural health monitoring (SHM) of bridges.

This paper was divided into nine sections. Section 2 will provide a comparison between some of the existing review and survey studies available related to NDE and the manner in which the present review provides recent and relevant insights for the researchers in the field of NDE for bridges. Section 3 discusses some of the salient features of the review methodology adopted in this review-based study. The Section 4 will outline a novel taxonomy that classifies the literature related to the NDE of civil infrastructure into three sub-sections, which will allow a better appreciation of the different ways in which the research in this area has evolved in the past years. A comprehensive discussion and evaluation of the different robotic platforms will be performed in Section 5. The discussion regarding the different sensory modalities for NDE-based data collection techniques will be provided in Section 6. Section 7 will highlight the different types of techniques developed for sub-surface- and surface-level analysis. Section 8 deals with the challenges facing the effective NDE of infrastructures in general and bridges in particular. Section 8 will also outline some of the limitations of the existing studies and potential recommendations in the field of NDE, specifically related to the three major themes within the proposed taxonomy. Section 9 will discuss the conclusion along with recommendations for future research in this research area.

## 2. Comparison with Existing Reviews

The purpose of this section is to provide a comparative evaluation between the different salient features of various existing surveys and reviews and the present review-based study with respect to examining the state-of-the-art in the field of NDE for bridges. In the past, a number of reviews have been published related to the field of NDE sensors and techniques [13] as well as some of the NDE data fusion techniques [14,15] and other aspects of research in the field of the NDE of civil infrastructure in general [16,17,18]. Table 2 [13,14,15,16,17,18,19,20,21,22,23,24] highlights the different review and survey studies that have been published in the recent past, along with their specific area of focus and the various limitations, which necessitate the need for renewing the understanding and knowledge of the state-of-the-art in the research area related to the NDE of civil infrastructure in general and bridges in particular. In order to assess the relevance of the information provided in earlier review studies, the most important factors include the time and scope of the evaluation. It can be seen in Table 2 that out of the total studies, nine have been published within the last five years [13,16,17,18,20,21,22,23,24], which means that the insights reported in these studies might still be relevant to the present research scenario. Two studies [14,19] were published more than ten years ago, which means that any findings examined are no longer relevant to the current state of the art. One of these studies [14] focused on the fusion techniques for NDE sensors. The other review-based study assessed the feasibility of the different algorithms for landmine detection using GPR sensors [19]. Another study [15] has a broad focus on multi-sensor data fusion in general for a diverse range of applications. Conversely, another one of the review papers [13] narrowly focuses on the studies related to one of the various NDE sensors, namely infrared thermography. The study reported by Chen [16] provided a quantitative analysis of the different studies reported in the field of construction automation. However, this study [16] did not provide an in-depth exploration of the salient features of the automation solutions and their applications for construction automation. Another relevant review study proposed by Agnisarman [17] focused on human-in-the-loop-based visual inspection systems for construction automation. There are a number of ways in which this study has a limited scope, as: (i) it does not focus on robot-based semi-autonomous and autonomous applications for construction automation, and (ii) this study only examined visual inspection-based sensors for SHM [17]. Similarly, another review-based study also provided an examination of vision-based applications for construction automation [18]. This study was limited in terms of its scope towards only identifying vision-based systems for construction automation [18]. The review of unmanned aerial systems (UAS) by another recent study only focused on the flying drones and the use of an infrared thermography sensor for building inspection [21]. Another review-based study dealing with non-destructive testing (NDT) methods for concrete bridges, which primarily focused on the different sensors for data collection for the NDE of bridges [20]. Five different types of sensors for the NDE of concrete structures were highlighted in another recent study [23]. Similarly, another review-based study discussed four types of non-contact testing sensors for the SHM of bridges, namely GPR, laser scanner, photogrammetry and infrared thermography [24]. However, these studies [20,23,24] only provide details regarding data collection techniques using a number of different NDE sensors.

In comparison with the prior reviews, this paper will differ in a number of ways. Table 3 highlights and compares the different research areas being examined by the different review studies that have already been discussed in Table 2. Collectively, the different topics explored by the existing review studies include the discussion related to various NDE sensors, the sensor fusion techniques in general for all applications and particularly for NDE and algorithms for SHM-related data analysis. 

In this review, the focus will be towards examining some of the recent technological developments in relation to the automation of NDE for SHM culminating in various single-sensor and multi-sensor systems with semi/full autonomous capabilities [8,12,16,17,18]. This review-based study will also cover some of the most important NDE sensors (e.g., visual sensors, impact-echo (IE), infrared (IR), ground-penetrating radar (GPR) and electrical resistivity (ER) sensors) and relevant studies utilizing data from aforementioned sensors towards the NDE of bridges in particular. For the effective SHM of bridges, there is a need to examine the integrity of the underlying structure using various data analysis techniques, which will also be discussed in this study. For the case of multi-sensor systems, this review will explore the varying sensor fusion techniques employed to ensure that bridge data from different sensors are used to provide a better assessment of SHM.

## 3. Methodology

In this section, the methodology for conducting the review performed in this research will be outlined. In order to ensure that only the most reliable and relevant studies were included within the review, there was a need to follow a specific set of guidelines that were proposed by a number of different studies in the past [25,26,27]. In this manner, researchers can attempt to replicate the review study successfully in the future. In the following discussion, some of the different elements of the review methodology will be discussed, which include the study design, search strategy, inclusion and exclusion criteria, selection process and data synthesis. Figure 3 outlines the way in which the quantitative assessment of the way in which the research papers were searched, selected, reviewed and short-listed throughout the different stages of the proposed research methodology. In the following sub-section, some of the salient features of the research study design implemented in this review-based study will be outlined.

### 3.1. Study Design

Before the details of the data collection process can be discussed, it is important to outline the different design elements of the research methodology implemented in this review-based study. The review-based study should be developed in order to minimize the risks of different biases as well as facilitate repeatability and transparency, especially when dealing with different types of data and analysis methodologies. A number of different studies have reported on different suitable methods for conducting a review of the relevant literature. Based on the meta-analyses of studies discussing the salient design-level features of review-based studies, Grant and Booth [27] have outlined 14 different types of review studies. In view of this classification, the present study can be termed as a ‘state-of-the-art review’. This type of review-based study not only acknowledges the findings from previous reviews, but attempts to update the information in light of the relevant research developments in recent decades that have not been addressed by existing reviews. Consequently, the *PRISMA* statement for reporting systematic reviews and meta-analyses [26] was used towards the selection and analysis of research papers related to the NDE of civil infrastructures and bridges in particular. In this respect, a number of different steps within the research methodology were shaped by the recommendations and best practices outlined by the *PRISMA* statement [26]. These different steps are outlined in the following sub-section of the paper.

### 3.2. Search Strategy

One of the authors (HA) devised a search strategy to extract the relevant literature from a number of online databases. The different reliable, peer-reviewed online databases searched for this review-based study included: *IEEE Xplore, Science Direct, Springer Link,* and *Google Scholar*. The keywords used to search for the relevant papers from the aforementioned databases included the following:For adding studies related to the non-destructive evaluation of bridges (autonomous/semi-autonomous robot-based and non-robot-based methods), the following keywords were used: ‘non-destructive evaluation of bridges’ and ‘NDE of bridges’;For adding studies related to the use of different sensors for NDE, some of the keywords used included ‘sensors for non-destructive evaluation of bridges’ and ‘NDE sensors for bridges’;For ensuring that a sufficiently broad-level examination of the relevant literature from the fields related to the NDE of civil infrastructure can be included in the early phases of the literature search and review, the following keywords were also used: ‘NDE sensors for civil infrastructures’, ‘non-destructive evaluation sensors of civil infrastructure’, ‘non-destructive evaluation for civil infrastructures’, and the ‘NDE for civil infrastructures’.

The scope of the keywords was intentionally selected to be sufficiently broad in nature. This allowed the extraction of all the relevant and important studies for the different research areas, along with some studies that were out of the scope of the present study. Such results could be easily filtered out in the remaining steps of the methodology, which will be discussed in the following sub-section. A number of different filtering options are available in the different online databases, which have been leveraged to ensure that only the most relevant studies are included in this review paper.

### 3.3. Inclusion and Exclusion Criteria

The purpose of the inclusion and exclusion criteria is to develop a set of standards that can allow the objective and effective evaluation of the different studies. In order to warrant inclusion in this review, the following set of criteria had to be satisfied: (i) the content of the study should be published in the English language, (ii) the study should deal with one or multiple aspects (e.g., platforms, sensors, or algorithms) of the NDE of infrastructure in general or bridges in particular, (iii) the study should have been published after 1 January 2000 to warrant inclusion in this review, (iv) the studies should be published in high-ranking peer-reviewed conferences and journals, and (v) the full version of the research paper should be accessible to the researchers. The exclusion criteria were used to ensure that the irrelevant studies that were included in the initial search results could be filtered out in a systematic fashion. Some of the exclusion criteria developed for this research are outlined as follows: (a) if the overall quality of the publication is sub-par (the use of erroneous language usage and/or the reporting of vague or inconclusive findings), the evaluation criteria are unverified and untested in nature and (b) the research focus of the paper does not fit into the scope of the review. Exceptions were made in the case of seminal studies and research works published as reports and books that were published before or after 2000. All of these included works are mentioned as additional sources within Figure 2. In order to practically implement the various inclusion and exclusion criteria within the online databases, filter options were available, which were used to ensure that the most relevant information was selected and included within the review-based study. In the following sub-section, some critical details of the selection process will be outlined.

### 3.4. Selection Process

In the earlier process of selection of relevant literature, the title and abstracts were separately screened by the two reviewers (HA and HML). The inclusion and exclusion criteria were applied to the individual articles by the two reviewers (HA and NG) on the randomly allocated set of articles, which led to the removal of various research articles during the screening process, as seen in Figure 3. In the later stage of the selection process, the full-text of the articles were reviewed independently by the two reviewers (HA and HML) to ensure the conformity of the individual articles to the specified exclusion and inclusion criteria outlined in the previous sub-section. In the final stage of the selection process, the remaining articles were discussed collectively by the two reviewers (HA and NG) in order to reduce bias in the selection of the individual articles. In the case of disagreement between the reviewers, Cohen’s Kappa was calculated [28]. The closer the value of Cohen’s Kappa to 1.0, the greater the consistency in the application of the inclusion/exclusion criteria between the two reviewers [28].

### 3.5. Data Synthesis

Due to the diverse nature of topics being covered, the study design and methodologies being followed, and the type of data analysis methods being used, it was challenging to perform a synthesis on the final set of studies obtained. In order to facilitate the evaluation of the findings from different studies, the overall selected set of literature was divided in terms of the taxonomy proposed in this research. Similarly, the overall analyses and subsequent findings were highlighted in a different manner. For each of the different literature streams, the data regarding the type of technologies, different sensors and algorithms were reported with variations in terms of the details. The details regarding the different literature streams investigated will be discussed in the following section.

## 4. Proposed Taxonomy

The NDE of civil infrastructure has been a widely discussed research area in the recent past. Figure 4 highlights the chronological developments in the research field of the NDE of civil infrastructure in terms of some of the major themes that are covered in this review. It can be seen from Figure 4 that the beginning period (1950–1978) of this research area focused on the development of novel sensing technologies, for bridge inspection in particular, and civil infrastructure in general [29,30,31,32,33,34]. Some of the major sensing technologies developed include GPR, infrared thermography, electric resistivity sensors, impact-echo-based techniques and ultrasonic pulse propagation-based methods. After that time period, the focus was devoted towards using different sensor fusion techniques and their application towards bridge inspection and civil infrastructure evaluation [35,36]. Some of the early studies related to robotics application towards bridge and civil infrastructure inspection were developed during the early 2000s. The development of semi and fully autonomous robotic systems expanded considerably after 2010 [37,38,39,40,41,42,43,44,45,46]. The different types of robots developed for bridge inspection can be broadly classified into ground, aerial and underwater robots. The complete details for each of these robot types will be explored in Section 5 of this paper.

In order to do adequate justice to the multi-faceted nature of the different tools, techniques, methodologies and technologies used in the prior studies, the proceeding discussion will be divided in the following manner:
Platforms: Different robotic platforms being used to assess the various physical characteristics of bridges. A taxonomy will be proposed, which will differentiate the different types of robot platforms for the NDE of bridges. Some of the essential components for evaluating the SHM of bridges will also be examined;Sensors: An array of instrumentation modules used for data collection will also be discussed with specific distinction between the single sensor-based and multiple sensor-based systems for NDE of bridges. Data from multiple sources require an additional level of complexity with regards to specifying the appropriate sensor fusion techniques. This particular aspect will also be analyzed in sufficient detail;Algorithms: A variety of techniques that facilitate the surface-level and the sub-surface-level structural evaluation of bridges will be highlighted in this section.

Figure 5 outlines the manner in which the aforementioned aspects of the research study are interlinked within the context of NDE of bridges. All of these elements are critical for the effective functioning of the robotic solutions. In this respect, the discussion in the following section will examine some of the different types of existing state-of-the-art technological robotic platforms for the NDE of bridges.

## 5. Technological Platforms

Traditionally, infrastructure evaluation has been considered a manual labor-intensive task, which is carried out by civil personnel using different sensors for data collection [8]. Despite recent technological advancements in this research area [8,37,47], a majority of the infrastructure evaluation is still performed by human operators using traditional modes for data collection, which are composed of standalone single sensor-based systems. In Figure 4, a better appreciation of the manner in which the available tools, techniques and platforms have evolved in recent years can be observed. It can be seen that there are fundamental divergences between the traditional methods and the innovative technological tools, techniques and platforms that have been employed for the NDE of civil infrastructure in the recent past. Most of the traditional tools utilize single-sensor-based systems, which means that the overall hardware and software requirements and complexities are limited in nature. However, most of the traditional tools and techniques require human operators a considerable number of man-hours to collect the data for assessing the structural fitness for a particular type of infrastructure. In contrast, the technologically advanced tools and techniques are efficient, such that they can use a limited amount of time to collect data from a wide array of sensors to provide an in-depth and multi-faceted assessment of the different structural deficiencies within infrastructures. In recent decades, there has been an increased focus towards the development and usage of semi-autonomous and fully autonomous robots for the NDE and SHM of civil infrastructures in general and bridges in particular. A wide array of diverse robots have been developed ranging from climbing robots (e.g., legged robots, wheel-based sliding robots and crawler robots) [38,39,40,48,49,50,51,52,53,54,55,56,57,58,59,60,61,62], and multi-rotor unmanned aerial vehicles (e.g., quad-rotors and octo-rotors) [63,64,65,66,67,68,69] to unmanned ground vehicles (UGVs) (e.g., advanced robotics and automation (ARA) lab robot, robotic crack inspection and mapping (ROCIM)*,* robotics-assisted bridge inspection tool (RABIT)) [45,47,70,71,72,73,74,75,76,77,78,79] and water-based robotic crafts (e.g., unmanned submersible vehicles (USVs), underwater marine vehicles (UMVs), underwater vehicles (UUVs)) [41,42,80].

Some of the recent studies have also focused towards developing hybrid robotic frameworks (e.g., wall-climbing unmanned aerial vehicles (UAVs), robots capable of flying and crawling and other multi-rotor flying robots capable of latching on to specific parts of infrastructure that require inspection), which are able to provide multi-functional roles and capabilities for the different types of inspection activities [43,81,82,83,84,85]. A number of different types of robots (e.g., flying robots, walking robots, sliding robots, climbing robots, and underwater diving robots) have been leveraged for the SHM and NDE of bridges in order to gain access to different parts of the bridges. For example, evaluating and inspecting tall steel beams above bridges can be a hazardous task for human inspectors to perform during different environmental conditions (e.g., rain, snow, wind, day and night conditions). It is for this reason that different types of climbing and aerial robots have been used to facilitate these tasks. In particular, the versatility of the aerial robots has allowed their increased utilization for the inspection of the different parts of bridges, such as the inaccessible underside of the bridge decks, higher parts of the bridge beams and cables [40,49,63,64,65,66,67,86]. Similarly, a number of different wheel-based and legged robots have also been used for inspecting concrete bridge decks, steel wires, concrete underside, and steel beams. 

A number of different robot platforms are designed for inspection activities for specific types of bridges (e.g., cantilever, arch, suspension, truss, cable-stayed, beam, girder and tied-arch bridges) [38,39,40,46,48,49,51,52,53,55,56,58,59,63,64,65,66,67,68,69,70,71,73,87]. In order to provide some level of insight regarding the different robotic solutions for bridge inspection, the proceeding discussion will focus on the taxonomy provided in Figure 6, namely: (i) ground robots, (ii) aerial robots, and (iii) marine robots. Some details regarding the different platforms are outlined in the following sub-section: 

### 5.1. Ground Robots

The majority of the robots developed for the SHM and NDE of bridges can be classified under the category of ground-based robots, in view of the taxonomy proposed in Figure 6. As it has been provided in Figure 6, the ground-based robots (these robots can also be termed as land-based robots, as they are developed to function on land) can be further classified based on the different types of locomotion capabilities developed, which allow them to inspect specific parts of the bridge infrastructure. *ROCIM* is a robotic platform that has been developed for bridge deck inspection [51,88]. Similarly, *RABIT* is another wheel-based ground robot with a wide array of sensors and autonomous navigational capabilities [12,40,47,49,51,70,79,88,89]. This particular robotic platform has been equipped with state-of-the-art sensor technologies (e.g., impact echo, ultrasonic surface waves, electrical resistivity and GPR), which enable the classification of some of the most common defects in bridge decks, such as concrete degradation, delamination and rebar corrosion [12,47]. The *ARA Lab Robot* is also a wheel-based robotic platform that has been recently developed for bridge deck inspection and maintenance [37,50,90,91]. With a similar array of sensors, another autonomous platform for infrastructural inspection was developed by La et al. [8], which provided a wide array of different functionalities related to the automated monitoring of civil infrastructure, using on-surface crack detection and bridge evaluation for signs of deterioration within the metal rebar and concrete slabs. In this particular research, the overall effectiveness of the automated robotic inspection system was also assessed for the evaluation of actual bridges [8]. A climbing robot was leveraged for the inspection of the underside of the bridge deck [59]. The majority of these robots have primarily been used for bridge deck inspection applications. A number of climbing, walking and crawling robots have also been developed, which are able to scale the vertical surfaces of the bridge infrastructure. Some of these robots include the BRIDGE (Bridge Risk Investigation Diagnostic Grouped Exploratory) bot [39], chain-like robot [52], magnetic wheeled robot [38], and the vortex climbing robot [69]. Most of the climbing and sliding robots dedicated to the inspection of different parts of the bridges are small-scale in nature, with a primary reliance on visual inspection methods using vision-based sensors.

Many bridges are equipped with cables to provide support and load balancing across the different parts of the bridge. To provide the automatic maintenance and inspection of these parts of the bridges, a considerable amount of studies focuses towards the mechanical design and development of different cable climbing robots [53,54,55,56,57,58,59,63,64]. The use of bipedal and quadruped legged robots has also been proposed for the inspection of civil infrastructures in general and the vertical structures of bridges in particular [87,92,93,94,95,96,97,98]. Table 4 summarizes some of the major characteristics of the different platforms for the NDE of bridges using a wide array of different robotic platforms and their respective sensory modalities. Due to the wide array of different sensors available for the NDE of bridges, the different sensory modalities are classified into *radars* (GPR sensors that employ EM waves of different frequency and wavelength), *vision* (all types of cameras and other sensors that provide visual information, e.g., red green blue (RGB), RGB-Depth (RGB-D), time-of-flight, thermal cameras, and sensors for infrared thermography), *acoustic* (all forms of sensors that employ sound for SHM, e.g., different IE methods and microphones, ultrasonic sensors) and *electric* (sensors that employ variations in current and voltage, e.g., ER, potential field mapping and Eddy current sensors) sensory modalities. When comparing the different ground-based platforms given in Table 4, it can be seen that the *RABIT* and *ARA Lab Robot* are the two robotic platforms with existing hardware and software capabilities that can allow them to function in an intelligent, autonomous fashion with regards to path planning, collision avoidance, trajectory planning, trajectory generation and sensor fusion techniques. As functional robots, they also have the ability to evolve over time, by equipping them with the state-of-the-art sensors for enhanced autonomous capabilities and data collection. In comparison, the majority of the other ground-based platforms relied on a single form of sensory modalities with limited hardware and software capabilities.

### 5.2. Aerial Robots

The recent breakthroughs in the field of aerial robots has allowed the usage of various multi-rotor platforms (e.g., four-rotor and eight-rotor-based platforms) in the field of SHM, with various implementation focusing towards bridge inspection and maintenance. The majority of the studies for bridge inspection using UAVs rely on visual inspection methods [72,73,74,75,76]. However, some of the recent studies have attempted to explore different ways in which aerial robots can be modified to provide perching and contact-based inspection capabilities [45,78,82,84]. A number of recent studies have also proposed the development of hybrid robots, which are able to provide multiple functionalities (e.g., flying and walking mechanisms and a number of different flying and contact-based approaches) [43,81,82]. Some of these platforms have provided a proof-of-concept with considerable potential towards successful utilization for bridge inspection in the future. Some of the different aerial robots deployed for the visual inspection of bridges are outlined in Table 4. For example, a UAV platform developed in [81] provided contact-based bridge inspection capabilities. Similarly, research by [45] examined the effect of contact force on pitch angle and vertical thrust force using one degree of freedom manipulator to perform the hammering analysis for bridge inspection. However, this platform did not rely on any NDE sensors, as the research is still in its initial stages. Another study focused on the position determination of UAV for the visual inspection of bridges using an on-board camera [76]. This is a relatively new field of research and further research is required in order to fully exploit the flexibility and versatility of aerial robotic platforms towards the accessing and monitoring of different parts of the bridge infrastructures.

### 5.3. Marine Robots

Marine robots primarily deal with the inspection of parts of the bridge infrastructure, which are submerged underwater. One of the earliest studies in this category emphasized the importance of examining and inspecting inaccessible or hard-to-access regions of the bridge infrastructure by human inspectors [80]. This platform made use of a camera for the visual inspection of submerged pier sections of the bridges [80]. However, the overall effectiveness of visual inspection is heavily affected by the clarity of the water and weather conditions, to name a few limitations of underwater standalone vision-based systems. Over the years, this area has expanded to receive attention with regards to post-disaster inspection as well as the regular SHM of bridge piers [42]. A number of unmanned marine vehicles (UMV), unmanned underwater vehicles (UUVs) and remotely operated vehicles (ROVs) have been deployed in the past, which include semi-autonomous sensory platform, *Muddy Waters*, *sea-RAI*, *VideoRay* and *YSI® Ecomapper* [41,42,80]. The majority of the limited number of robotic platforms deployed underwater rely on the visual sensory information for assessing the SHM of a bridge structure submerged under water, as it can be seen from Table 4. However, due to the various challenges associated with underwater inspection, there is a need for further research, which can provide improved sensory capabilities for data collection as well as tools and techniques for analyses, which can be used for the underwater SHM of bridges in the future. At the same time, there is also a need for performing the comprehensive feasibility of the developed and deployed robotic platforms within different underwater conditions for the SHM of different bridges. In the following section, the prime focus will be towards discussing some of the different algorithms and techniques developed for the NDE of bridges.

## 6. Tools and Techniques for Data Collection

Sensors allow the different NDE platforms to collect data, which can be used to assess the overall conditions of the infrastructure in terms of its suitability and safety for humans in the near future without endangering their lives in any way. For studies in this particular field, it is important to incorporate sensors, which are able to analyze the overall internal and external conditions without physically tampering with the infrastructural materials (e.g., concrete, steel). In the previous section, it can be seen in Table 4 that the different types of sensors deployed on the various platforms had been classified into four main categories, namely vision, acoustics, radar, and electric sensors. The classification proposed in Table 4 was based on the different sensory modalities that were equipped in the different NDE platforms. However, in this section, the scope is broader than the usability and applicability of the variety of sensors on NDE-based robotic platforms. In this section, the primary discussion will relate to the different types of sensor-based systems utilized for infrastructure evaluation and structural health monitoring. It can be seen in Figure 7 that NDE systems can be broadly classified into *single-sensor* and *multi-sensor*-based systems. Most of the single sensor systems, such as *Roadmap* [102] and *BYU (Brigham Young University) IE scanner* [103] employ a single sensor each, namely GPR and IE respectively. Conversely, multi-sensor systems (e.g., *RABIT* [12,40,47,51,70] and *Seekur Jr.* [8]) make use of different types of sensors, which allows them to provide an accurate and multi-faceted evaluation of the infrastructure, such that the limitations of one sensor type (e.g., infrared thermography, which is limited in terms of providing information regarding near-surface delamination [89] can be mitigated by the use of other sensors (e.g., GPR, which is able to provide information regarding the structural defects present at sufficient depth underground [47]). However, this platform did not rely on any NDE sensors, as the research is still in its initial stages. 

Another study focused on the position determination of UAV for the visual inspection of bridges using an on-board camera [76]. This is a relatively new field of research and further research is required in order to fully exploit the flexibility and versatility of aerial robotic platforms towards the accessing and monitoring of different parts of the bridge infrastructures. In Figure 7, each single sensor-based system leverages one type of sensor (e.g., sensor *a*, sensor *b* and so on). Therefore, the discussion within single-sensor systems will deal with the different types of sensors used for the NDE of bridges. For the case of multi-sensor systems (*A*, *B*, … *N*), such that each system is composed of *n* sensors (*A_1_*, *A_2_*, … *A_n_* for system *A*; *B_1_*, *B_2_*, … *B_n_* for system B and so on), such that *n* ∈ *N*. It can be seen in Figure 7 that in order for the multi-sensor systems to extract meaningful insights from multi-sensor data, there is a need for implementing sensor fusion algorithms (fusion algorithm *F_a_* for system *A*, algorithm *F_b_* for system *B* and so on). Therefore, the primary focus of the discussion within the multi-sensor systems will deal with the rationale behind the utilization of sensor fusion and the different algorithms proposed in the previous studies. The following sub-section will discuss some of the available sensory modalities for the NDE of bridges, which are widely utilized within single-sensor-based NDE systems.

### 6.1. Single Sensor Systems

It has already been discussed in the prior sections that many of the earlier studies in the field of the NDE of bridges deal with single sensor-based systems. At the same time, according to Table 4, the majority of the existing robotic solutions developed can also be classified as single-sensor-based systems. However, there are a number of different sensory modalities available for the NDE of bridges and civil infrastructures in general, which will be discussed in sufficient detail in this section. For the case of impact-echo-based NDE techniques, metallic objects (e.g., metal chains, ball bearings) are used to create acoustic vibrations and contact sensors are employed to record the reflected sound waves from the different underground materials and the defects within infrastructures. A number of studies in the field of the NDE of infrastructures have reported the utilization of impact-echo sensors in the recent past [103,104,105,106,107,108,109,110,111]. These studies have focused towards examining the different types of defects present within civil infrastructures. For the case of the research proposed by Zhu and Popovics [105], an air-coupled impact-echo sensor and recording devices have been used to analyze the extent and depth of delamination within concrete structures. The effectiveness of the impact-echo sensors is dependent on the type of impactor used [105]. This particular tool for data collection has been extensively used for infrastructure evaluation in a number of recent studies [103,108,112]. Another study validating the effectiveness and efficiency of air-coupled sensors was also validated [107]. However, for rapid scanning-based applications, the use of air-coupled sensors can pose challenges for real-time data collection [108]. To improve the overall efficiency of data collection for NDE-based applications, different studies have developed automated IE systems with complicated electromechanical mechanisms for the repeated and consistent impacts on structure surfaces [103,108,113,114]. Figure 8 provides details regarding the impact-echo method and its utilization for the NDE of civil infrastructures using different components, such as *impactors* (metal objects used to create sound vibrations), *transducers* (sensor used to detect the sound reflections), *data acquisition module* (hardware components used to filter sound vibrations) and the *data analysis module* (software used for analyzing and visualizing the signal output from the sensor over time). 

In the wake of technological improvements in commercially available infrared detectors, infrared (IR) thermography has gained considerable popularity in the past decade, specifically after being established as an American Society of Testing and Materials (ASTM)-certified method to detect delamination in bridges in 2003 [13,115]. The use of infrared thermography has been discussed, not only in the context of infrastructure evaluation [13,116,117,118,119], but also for tunnel excavation [120,121,122] and for the examination of different materials (e.g., metals, aluminum laminates, carbon fiber-reinforced polymers and glass fiber-reinforced polymers), which is important towards assessing the structural integrity of different mechanical parts specifically developed for the aerospace industry [123,124]. In comparison with other methods for the NDE of infrastructure, infrared thermography has been recommended to provide the real-time, objective assessment of infrastructural health, specifically for the case of near-surface delamination detection [125,126,127]. However, with the increasing depth of the underground defects within infrastructures, the overall accuracy and reliability of the infrared thermography technique decreases substantially [128,129,130]. Some of the other factors which can impede on the accuracy of the data collected include the type of sensor equipment being used, shadows, moisture, surface debris, wind speed and sustained solar heat [118,131,132].

The governing principles underlying the usage of infrared thermography include conduction, convection and radiation. For the case of the near-surface delamination of the presence of underground void spaces, infrared thermography leverages the concept of variation of the temperature gradient between the defective and non-defective regions within infrastructures [133,134]. Figure 9a,b outline the ways in which heat conduction and radiation emission during day and night time, respectively, allow the infrared thermography sensors to differentiate between delaminated and non-delaminated regions [133]. It can be seen in Figure 9c,d that the regions with potential underground structural defects are visible as brighter regions on the thermogram [134].

Another widely utilized sensor is the ground-penetrating radar (GPR), which has been the focus of a number of recent studies related to the infrastructure evaluation and SHM of civil infrastructures and bridges [8,37,47,135,136,137,138,139,140,141,142,143,144,145,146]. Within civil engineering, GPR has been utilized for diverse applications, which include, but are not limited to, detecting and measuring pipes, mines, other underground utilities, the health monitoring of bridges, railway tracks, tunnels, roads and pavements, as well as for the detection and localization of underground rebar [8,37,47,136,137,138,139,140,141,142,143,144,145,146,147,148]. The B-scan data from GPR sensors provide a visual transformation of the radar waves reflected from different parts of the underground infrastructure (e.g., concrete, steel rebars, void spaces), which can be used to highlight the corrosion, delamination, presence of void spaces and structural damage to rebars [37,135,137,138]. The details regarding different data analysis techniques will be discussed in the following sub-section. Figure 10a provides information regarding the underlying principles for wave propagation using the fixed-offset-based method for data collection using GPR [149,150]. Similarly, the use of a common midpoint-based method has been shown in Figure 10b [149,150]. The radar waves from the transmitter penetrate the ground, and based on the different properties of the underlying construction materials and other artefacts (e.g., location, dimensions, density, depth), the intensity and signature of the waves reflected back from the different regions to the receiver can vary to a considerable extent.

Vision-based sensors have received considerable attention in the recent past towards the NDE of diverse civil infrastructures, ranging from sewers [151], tunnels [152,153], structural ceilings [154], roads [155,156,157], dams [158], pavements [159,160,161,162], and bridge decks [40,48,49,163,164]. The advent of state-of-the-art learning-based techniques for data analysis has facilitated the widespread usage of vision-based sensors within different robotic systems for the NDE of bridges [12,37,40,47,49,50,51,70,79,88,89,90,91,99]. The visual inspection of infrastructure is important to provide information regarding the surface-level defects and damages in concrete. A number of vision sensors have been utilized to perform the SHM of civil infrastructures, namely the smartphone camera [154], digital cameras [8,88,89,99,152,165], depth sensors [42,164], time-of-flight cameras [166], closed-circuit television (CCTV) [151], laser-scanners [77,98,157,164] and Visual SONAR (Sound Navigation and Ranging) sensor [41]. The vision-based sensors are dependent on the reflection of light from infrastructure surfaces to provide an assessment of surface-level NDE. Based on the data in Table 4, it can be seen that many of the aerial robots are equipped with different visual sensors to provide surface-level information to assess structural health of bridges [72,74,75,76,77,83,98].

A number of different electrical sensors have also been used for the assessment of SHM of infrastructures [12,37,40,47,49,50,51,52,59,70,79,88,89,90,91,99,100,101], which are primarily used in a ground-based robot. A half-cell potential sensor was used by the ETH (Eidgenössische Technische Hochschule) Zurich autonomous robot for potential mapping to detect a level of corrosion within the concrete structures (e.g., bridge deck underside and parking lots) [59]. Electrical Resistivity (ER) probes have been one of the most widely used electrical sensors, which have been incorporated within two of the most widely discussed ground robots for the SHM of bridge decks in the recent past, namely the *ARA Lab Robot* [37,50,90] and *RABIT* platforms [8,91,100,101]. The purpose of ER probes is to examine the level of sub-surface corrosion within bridge decks and other infrastructures [79]. The *RABIT* platform is equipped with four Wenner-type ER probes; two outer probes generate an electrical current and the two inner probes measure the intensity of electrical field, which is used to calculate the electrical resistivity [79]. Another type of electrical sensor was used by the steel climbing robot, namely the Eddy current sensor [53,100,101], which is used to measure the level of corrosion, rust and crack within the steel structures of bridges.

### 6.2. Multi-Sensor Systems

For the case of traditional tools, techniques and platforms, any one of the aforementioned data collection methods (e.g., impact echo, GPR or infrared thermography) can be utilized for performing SHM and the assessment of civil infrastructures. However, state-of-the-art platforms (e.g., *Seekur Jr.*, *RABIT*) [8,12,40,47,51,70] have utilized an array of different sensors, which provide an in-depth and holistic evaluation of the civil infrastructures. In this section, the primary purpose is to explore the different ways in which sensor fusion techniques can be used for different modalities (discussed in the previous section) to collectively allow the development of efficient and cost-effective systems for the NDE of bridges.

Sensor fusion within multi-sensor systems allows the improvement in the ability of those systems towards effectively obtaining insights from the available data. Prior studies have revealed that sensor fusion techniques also improve the overall accuracy and efficiency as well as reduce the data-level and system-level redundancy of multi-sensor-based systems [167,168]. Sensor fusion is being widely utilized in a wide array of applications, ranging from medical diagnostics [169], aerospace [170], plant inspection, and high-precision manufacturing [171,172] to remote sensing [173] and the NDE of civil infrastructures [37,168]. One of the various functions of sensor fusion is to ensure that the data from multiple sources can be combined together to facilitate appropriate analyses, leading to the enhanced efficiency and effectiveness of the deployed practical systems [174,175]. Some of the earlier studies have explored different types of fusion techniques, which can be broadly classified into the following:Data-level fusion: Raw data from the different sensors are transformed and concatenated together. A single technique for data processing and analysis is applied collectively to the fused data from the different sensors [14,128,176];Feature-level fusion: Feature from the multi-modal data are collated collectively. In order to ensure that data from different sensors are fused together effectively, different types of data transformation technique are utilized [14,128,176];Classifier-level fusion: A number of different classifiers are used together to develop hybrid classifiers. The final performance of the hybrid classifier is based on the average of the individual classifiers chosen for analyzing multi-sensory data [14,128,176];Result-level fusion: A number of techniques are employed to individually analyze data from individual sensors. The results from each method are combined together based on specified criteria [14,128,176].

It is important to understand that different sources provide varying classifications for understanding the different types of data fusion techniques. For example, a review of NDE sensor fusion techniques by Liu et al. [14] outlined the different approaches that can be broadly divided into signal-level, feature-level, pixel-level and symbol-level fusion techniques. Meanwhile, the survey of different data fusion techniques in another study developed a taxonomy in terms of the different underlying challenges, namely data imperfection, data correlation, data inconsistency and varying data forms [15].

For catering to the various technical requirements of multi-sensor-based systems, a major stream of literature related to NDE has focused towards developing the different techniques of sensor fusion [14,37,128,168,177]. A common method for data fusion is the Dempster–Shafer Theory, which has been widely used for NDE applications ranging from data fusion between ultrasonic and X-ray imagery to the techniques developed from the data fusion between infrared thermography and GPR-based sensor modalities [167,168,178]. One of the earlier studies on the fusion of data from NDE sensors (e.g., GPR, portable seismic analyzer, and falling weight deflectometer) utilized a number of different fusion techniques, which include fuzzy logic, Bayesian, statistical weighted average and hybrid fusion techniques [179]. The use of adaptive fusion operators with customized decision criteria was employed by one of the relevant studies for performing the fusion of multi-modal data (e.g., electrical resistivity, ultrasonic waves, infrared thermography) represented in the form of possibility distributions and fuzzy sets [168]. The use of radar and ultrasonic sensory modalities was performed in another study by Maierhofer and colleagues [167], which utilized data pre-processing, normalization, filtering and amplification techniques for data transformation before different fusion techniques (e.g., summation, subtraction and maximum amplitude-based methods) were performed on the multi-sensor data.

In their research, Kee et al. [128] proposed the usage of tools related to infrared thermography and air-coupled impact echo, along with their fusion techniques, which was based on simple rule-based criteria to provide valuable insights towards effective infrastructural maintenance. However, the data were not obtained from an actual bridge, but from a bridge test-bed specimen, which means that this method has not been tested on a practical system using data from actual bridges. Data from a wide array of different sensors were utilized within the *RABIT* platform in another study related to the NDE of bridges [8]. However, the use of sensor fusion techniques was not outlined to ensure that the data from different modalities could be effectively leveraged to provide a holistic evaluation of the SHM of bridges. For the case of the deployment of the *ARA Lab Robot* for the NDE of bridges, a sensor fusion algorithm was deployed, which allowed the robot to optimize the duration of time taken to perform the necessary operations required for data collection using a camera, GPR and IE sensors across the different regions of the bridge deck infrastructure [37]. According to Brierley et al. [172], the questions related to the selection of an appropriate sensor fusion technique is application specific in nature. Some of the considerations in this regard are given as follows: (i) the type of problem being addressed, (ii) the type and scale of the sensory data being handled, and (iii) the assessment criteria for performance evaluation, as the sensor fusion techniques can provide varying results in different contexts and applications [172].

## 7. Algorithms for Data Analysis

In this section, the focus will be towards outlining the different analysis techniques using data from a wide array of NDE sensors for bridge infrastructural evaluation, which were highlighted in the previous sub-sections. The discussion in this section will be divided into two sub-sections, namely: (i) different techniques pertaining to analysis of the surface-level data for bridges and other civil infrastructures, and (ii) myriad of techniques developed for analyzing sub-surface-level data for bridge decks. The surface-level analysis for NDE is used to examine the level of cracks, and the corrosion of the concrete surfaces of civil infrastructures. The sub-surface-level analysis allows the evaluation of the level of cracks, corrosion and delamination within bridge decks.

### 7.1. Surface-Level Analysis: Concrete Crack Detection

There is a considerable amount of research effort devoted towards concrete crack detection for the surface-level SHM of civil infrastructures in general and bridges in particular [37,48,51,88,163,180,181,182,183,184,185,186,187,188,189,190,191,192,193,194,195,196,197,198,199,200,201,202,203,204,205,206,207,208,209,210,211,212]. Due to the concrete-based composition of the majority of civil infrastructures (e.g., dams, roads, buildings, sewers, bridges, tunnels), the techniques developed for concrete crack detection within a particular type of concrete structure can also be generalized towards other civil infrastructures. Some of the earlier works focused on the utilization of basic-level image-processing techniques for crack detection in concrete structures [48,51,88], which included basic-level morphological approaches [166,190,191,192,193], digital image correlation techniques [194,195,196,197] and different segmentation-based approaches [198]. A number of different image-based filtering techniques were also employed, namely Gabor filtering [199], median filtering [184,200], texture filtering [201,202] and data fusion-based filtering approaches [176,203]. The efficacy of different image transformation techniques has also been discussed, ranging from the watershed transform [190,192,194], wavelet transform [200,201,204,205], and randomized Hough transform [200,206]. The use of fast Fourier and fast Haar algorithms with Sobel and Canny edge detectors was developed for the concrete crack detection in one of the earlier studies [207]. A block-based crack detection approach was developed for the bridge decks in another study [37]. Another study made use of histogram-based method for the extraction of crack features from the input images of bridge decks [48]. A genetic learning-based network optimization algorithm was also proposed with the application for concrete crack detection [181]. For the classification of images into crack and non-crack regions, the use of Support Vector Machine (SVM) was proposed for effectively detecting and classifying the cracks in bridges [187]. Data from the GPR sensor were used for sensitive concrete crack detection using a super high-frequency band system and time-variant deconvolution-based approach [208].

The use of different deep-learning frameworks has gained considerable attention in recent works related to concrete crack detection [157,180,182,189,209]. A deep-learning-based SSD Inception V2 and SSD MobileNet models for concrete road damage detection was developed in another recent study [155,159,210]. Similarly, the crack detection problem was solved using semantic segmentation with the help of deep residual neural networks (NNs) in a number of recent studies [211,212]. A Faster-region-based CNN model (Faster R-CNN) was proposed towards the quasi-real-time system development for the detection of different types of defects (e.g., concrete crack detection, steel delamination, bolt corrosion, etc.) [152,209]. Another recent study utilized a U-net-based fully connected CNN model for concrete crack detection [188]. Some of the most recent studies have made use of different encoder–decoder-based deep-learning architectures to improve the existing limitations of crack detection systems using a pixel-wise classification of concrete images [182,189]. [213,214,215]. *DeepCrack* is the name proposed for a deep learning-based framework designed specifically for the crack detection using crack probability maps obtained with the help of a deep-encoder–decoder-based network [216]. A comparison in the performance and techniques employed by the different studies in the past towards concrete crack detection have been outlined in Table 5. A number of other deep-learning frameworks have also been developed for crack detection, such as the *CrackNet* [217], *CrackNet II* [218] and *CrackNet V* [219] models, which have further improved the performance of the crack detection systems. Another study made use of multiple visual sensors (e.g., digital camera, laser scanner and distance sensor) for concrete crack detection and measurement with the help of the YOLO (You Only Look Once)-v3-tiny model [164]. An autonomous crack width-measurement system using medial axis transform and flexible kernel was also proposed in another recent study [220].

### 7.2. Sub-Surface-Level Analysis: Rebar Detection and Localization

One of the primary emphases of this section will be towards the algorithms proposed for rebar detection and localization, which is essential for the structural health monitoring of bridges. For the case of bridge monitoring, earlier studies utilized different pattern recognition and image-processing techniques for rebar detection and localization by hyperbola extraction from GPR radargrams [221,222,223,224]. Different hand-crafted features were employed by many of the earlier studies, such as the edge-based features [146,147,148,225,226,227], texture-based features [222,228], template matching [221,229], histogram of oriented gradients (HOG) [47,135,140], feature transformation methods, e.g., Radon transform [142], and Hough transform [230,231,232,233] and statistics-based methods, such as clustering-based approaches [223,234], least-square methods [235], and higher-order moments [235,236]. These features were trained using a wide range of different learning-based techniques towards developing effective rebar detection and localization algorithms in the past [47,135,136,137,138,227,237]. Research by Gibb and La [135] trained a Naïve Bayesian classifier using HOG features. Support Vector Machine (SVM) has also been used in prior studies [47,145]. A number of different neural network (NN) frameworks have also been employed in earlier studies for rebar classification [147,148,227,237]. Many of the earlier methods failed to effectively leverage the capabilities of NN models using edge-based features [147,148,227], which are not suitable for real-world systems dealing with GPR data that contain varying rebar signatures and fluctuating noise levels. Some of the recent studies have made use of convolutional neural networks for rebar detection [136,137,138,238]. A study by Dinh et al. [136] proposed the usage of a 24-layer deep CNN model for rebar classification. The use of residual neural networks (ResNet-50) has also been proposed in recent studies related to rebar detection and localization [137,138]. The preliminary examination of the results using GPR data from real-world bridges has shown that different ResNet models (i.e., ResNet-18, ResNet-32, ResNet-50, ResNet-101, ResNet-152) provide increased accuracy and generalizability [137,138,239]. Another study implemented the multi-objective genetic algorithm for the classification of rebar images [240].

Due to its critical importance, many of the studies focused on the development of rebar detection and localization systems in a collective fashion [47,135,136,138]. The earlier studies made use of hand-crafted features with edge-fitting or curve-fitting algorithms to localize rebar signatures from GPR radargrams containing multiple rebar profiles [143,147,221,224]. Hough transform fitting has also been leveraged with edge features to localize individual rebar profiles [224]. Yuan et al. [226] proposed the drop-flow algorithm using edge features to decompose individual hyperbolas and cater to over-segmentation. The edge-feature-based localization methods suffer from a lack of generalizability to rebar size, dimensions, location and variations in the noise levels. An expectation-maximization algorithm was proposed by Chen and Cohn [235], which has various limitations for implementation in real-time systems, in terms of computational complexity, difficulty in convergence and sensitivity to the variations in configuration points. A column-connecting clustering algorithm with orthogonal hyperbola fitting was developed in [139]. Another study proposed a precise hyperbola localization algorithm [135], which made use of the hyperbola fitting and local maxima. However, this method has limitations towards providing results for real-time systems. In contrast to previous hyperbola-fitting methods, the study by Kaur et al. [47] made use of random sample consensus (RANSAC), which is an iterative method for robust hyperbola fitting with corrective capabilities, specifically with noisy data and outliers.

The different characteristics of the studies related to rebar detection and localization have been outlined in Table 6. There are various other studies utilizing GPR sensors with rebar detection and localization algorithms, which focus on other underground buried objects, such as landmines, void spaces, and pipes. These studies have not been included in the table, as they are beyond the scope of this discussion. Despite considerable research in this field, the effective acquisition of hyperbolic signature remains a complicated research problem with various challenges, such as the separation of intersecting rebar profiles, the full/partial occlusion of hyperbolic signatures, and the complexities within the underground spatial configurations [139,226,239].

## 8. Challenges

There are a number of different challenges affecting the development of autonomous/semi-autonomous systems for the NDE of civil infrastructures in general and bridges in particular. Most of the processes underlying NDE systems (e.g., manufacturing NDE, industrial NDE and civil infrastructure NDE) are time and resource intensive in nature. This means that NDE is performed only if the cost of failure (in terms of capital and loss of human lives) is greater than the costs associated with performing NDE, but with timely inspection and remedial measures, the probability of defect can be substantially reduced [172]. Over the years, a number of robotic platforms have been developed for performing the NDE of infrastructure [8,12,37,47,52]. However, there are a limited number of different initiatives towards the development of semi-autonomous/fully autonomous systems that can reduce the overall time and resources taken to provide regular health monitoring services for civil infrastructures. It is for this reason that the current situation warrants the development of robust, replicable and cost-effective technological platforms for the SHM of civil infrastructure studies.

From a sensor and data fusion perspective, there are a number of challenges that have hindered the development of effective sensor fusion techniques for practical systems in the past. The deployment of contact-based sensors for the NDE of civil infrastructure is time-consuming in nature [37]. In this respect, there is a need for the development of stochastic, optimal path planning algorithms that utilize sensor fusion-based decision making to differentiate between areas of higher and lower priorities for inspection, depending on the input from different sensory modalities. At the same time, the majority of the existing studies related to sensor fusion do not provide reliable performance evaluation metrics [15]. Some of the other challenges towards the development of performance criteria for effective sensor fusion techniques include a lack of effective ground truth, multiple, often conflicting dimensions of different performance metrics, and the need for the modification of the performance criteria for sensor and data fusion in view of the underlying criteria, context and applications [15]. A similar issue has also been encountered with respect to the examination of studies related to rebar detection and localization. In Table 6, only a handful of studies have been reviewed, as the majority of the studies related to rebar detection and localization do not provide reliable and effective performance evaluation metrics. Apart from that, there are many studies that do not provide any information regarding the performance criteria used to assess the quality of findings in their respective studies.

Many of the existing studies utilizing robotic platforms for the SHM of bridges rely on single sensors, which can provide limited insight into the multi-faceted problem related to the SHM of bridges and civil infrastructures in general. In this regard, the increased utilization of vision-based inspection methods has been stressed in one of the recent studies [18]. A scientometric analysis of the relevant research field provided insight regarding the lack of multi-disciplinary research, which is hindering and limiting the development of effective solutions for the NDE of bridges [18]. A number of studies in other research domains related to robotics have extensively examined the human-level factors that affect the development of relevant systems. However, the use of civil infrastructure-based robotic systems has not examined the human factors, which include, but are not limited to, the civil inspectors’ skills, trust in automation-based platforms, situation awareness, and the workload demands of civil inspectors [17]. There is also a need to assess the human–robot collaborative factors that can determine the effectiveness of deploying inspection-based robotic platforms in the different environments and contexts within the field of civil infrastructure evaluation.

Another limitation towards the development of effective real-time robotic solutions for the NDE of bridges is the lack of adequate funding towards the development of automated solutions to provide the regular SHM of bridges. In order to expedite the process of the regular maintenance of public infrastructure, there is a need for investment in the field of NDE of infrastructure, which can allow the development of effective autonomous and semi-autonomous platforms. One of the recent studies in this regard emphasized the high costs regarding development, the testing and practical deployment of on-ground robotic systems for facilitating the SHM of civil infrastructure [16]. Earlier studies have emphasized the effectiveness and superiority of the robot-based NDE systems in comparison with the traditional maintenance techniques that have been in practice in the past [16,135]. Nevertheless, the overall advantages of developing autonomous/semi-autonomous robotic platforms for SHM outweigh the underlying costs and challenges. The next section (which will be the final section of this paper) will conclude some of the major elements of the discussion that have been highlighted in the different sections of this review-based study, along with recommendations for future studies in the field of the NDE for bridges.

## 9. Conclusions and Future Works

This paper has provided a comprehensive review of the state-of-the-art robotic platforms, sensors and algorithms that have been developed for bridge inspection and evaluation with considerable implications for civil infrastructures in general. There are a number of different relevant topics that have been addressed in this review paper. In order to effectively highlight the novelty of this review-based study, a cross-level comparison between different reviews and surveys was discussed. The level of difference in the focus and scope between the existing review-based studies has also been examined. Unlike prior review-based studies, this review attempts to provide a holistic overview of the relevant research area by highlighting the three critical themes of the robotic systems for the SHM of bridges, namely the robotic platforms, sensory modalities and data analysis techniques. This review also provides a detailed evaluation of the different challenges, as many of the aspects highlighted in that section have not been adequately covered by existing reviews in the past. The novel contributions of this paper can be highlighted in terms of the manner in which the relevant literature has been structured in the perspective of three major inter-linked themes outlined in Figure 5. One of the major themes that has been discussed in this review is related to the different *robotic platforms* that have been developed for the NDE of bridges. A taxonomy of the different robotic platforms has been presented along with some of the salient features of the different robotic systems that have been developed in recent studies. It has been highlighted that the different robot systems can be broadly classified into single-sensor and multi-sensor-based systems. Therefore, the second theme examined the different sensors that have been used for the NDE of bridges. For the case of single-sensor-based systems, the focus has been towards the individual types of sensory modalities (e.g., GPR, IE, ER, vision). For the case of multi-sensor-based systems, the focus has been towards exploring the different sensor fusion techniques leveraged for the SHM of bridges and civil infrastructures in general. The third and final theme that has been highlighted in this review focused on the different types of algorithms developed for the *surface-level* (concrete crack detection) and *sub-surface level* of analysis (rebar detection and localization). In the later parts of the review-based study, some of the major challenges have also been examined in view of the development of effective autonomous robotic systems for the real-time NDE of civil infrastructures in general and bridges in particular.

There are a number of different ways in which future studies can attempt to extend the state of the art in the SHM of bridges and civil infrastructures. The use of multiple sensors should be explored in future studies for the development of solutions for the SHM of bridges and civil infrastructure [18]. The use of multiple sensory modalities will allow the examination of the various aspects of the infrastructural deficiencies within the different civil infrastructures and bridges in particular. The previous section highlighted a number of different challenges being faced towards the development of robotic solutions for the NDE of bridges in particular and civil infrastructure in general. Future researchers can work towards developing effective strategies for managing or mitigating the different challenges. At the same time, the emphasis of relevant studies in the future should be demoted towards the development of state-of-the-art robotic platforms, in particular the aerial and underwater platforms, which currently lack in terms of practical deployment and effectiveness in practical scenarios. Therefore, future studies should primarily focus towards the practical effectiveness of the aerial and underwater robotic platforms with practical testing and evaluation on actual civil infrastructures.

## Figures and Tables

**Figure 1 sensors-20-03954-f001:**
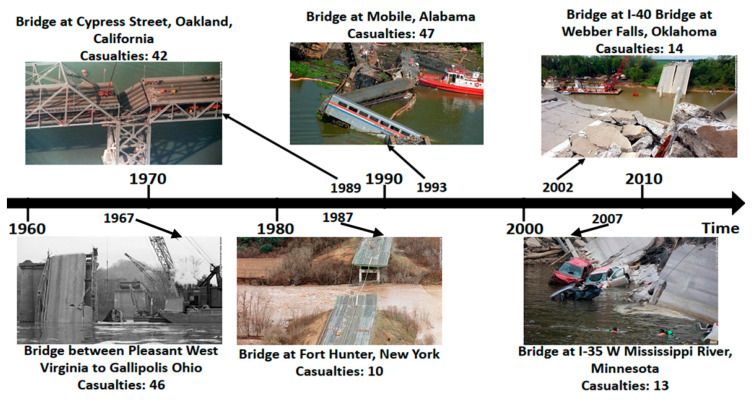
Timeline showing some of the bridge accidents in the U.S. in recent decades [1].

**Figure 2 sensors-20-03954-f002:**
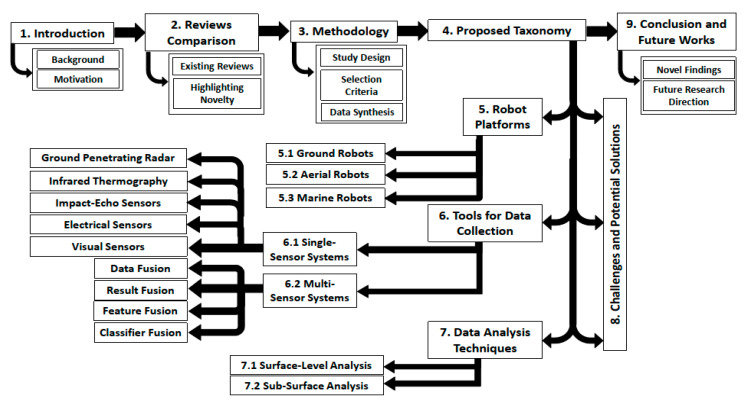
Roadmap of the review-based study in order to understand the inter-relationship between the different sections of the paper.

**Figure 3 sensors-20-03954-f003:**
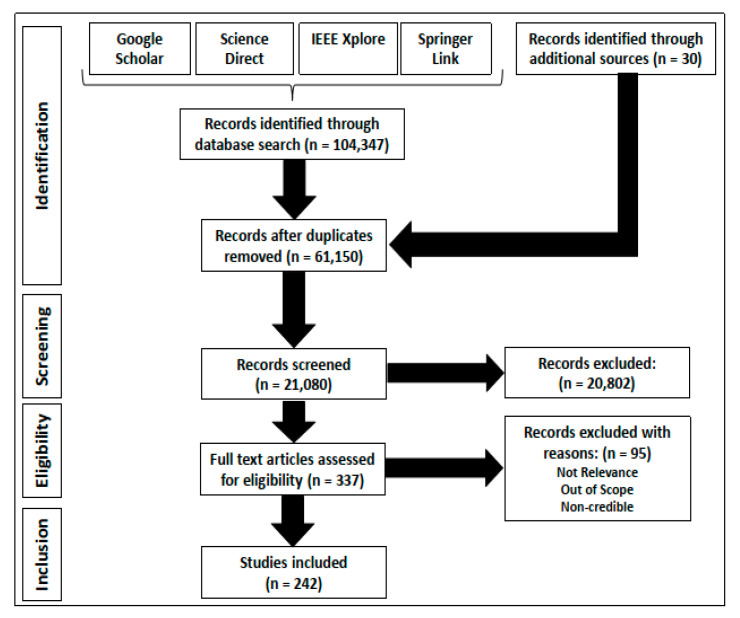
The different steps in the review process for the evaluation of the relevant literature.

**Figure 4 sensors-20-03954-f004:**
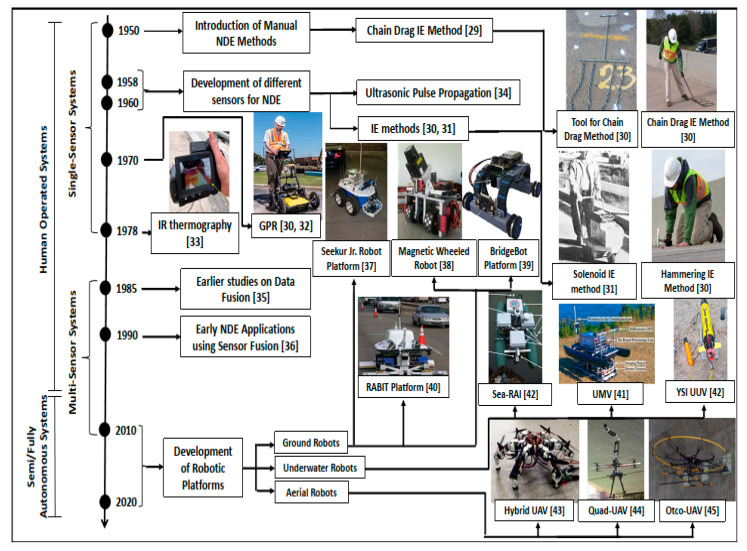
Timeline depicting the applications of the different sensors and the development of different platforms for NDE.

**Figure 5 sensors-20-03954-f005:**
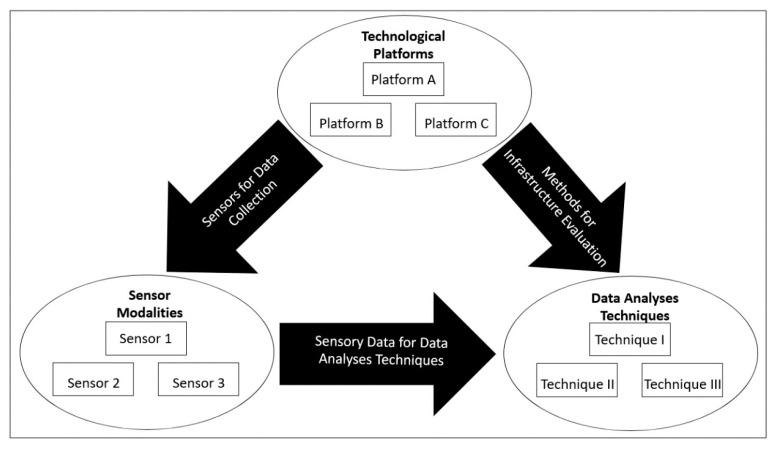
Visual representation for the proposed taxonomy used for studies on the NDE for civil infrastructure evaluation.

**Figure 6 sensors-20-03954-f006:**
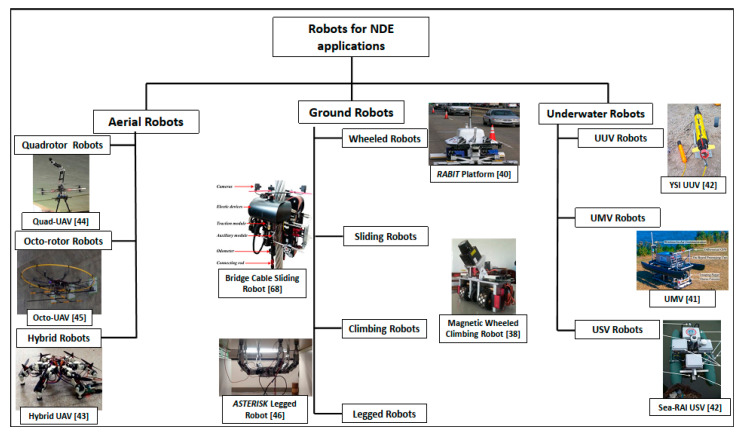
Proposed taxonomy for NDE robot development for bridge inspection.

**Figure 7 sensors-20-03954-f007:**
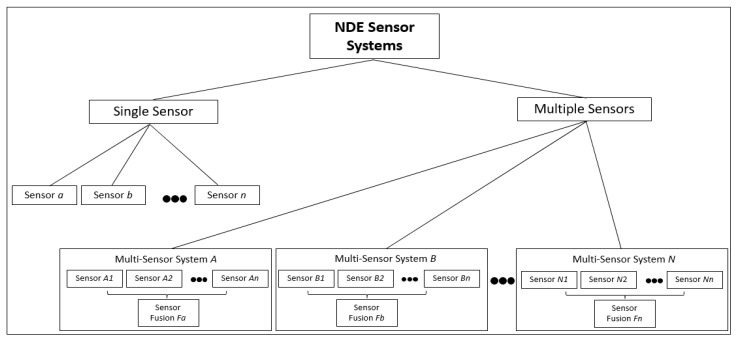
The different sensor configurations within the NDE evaluation literature.

**Figure 8 sensors-20-03954-f008:**
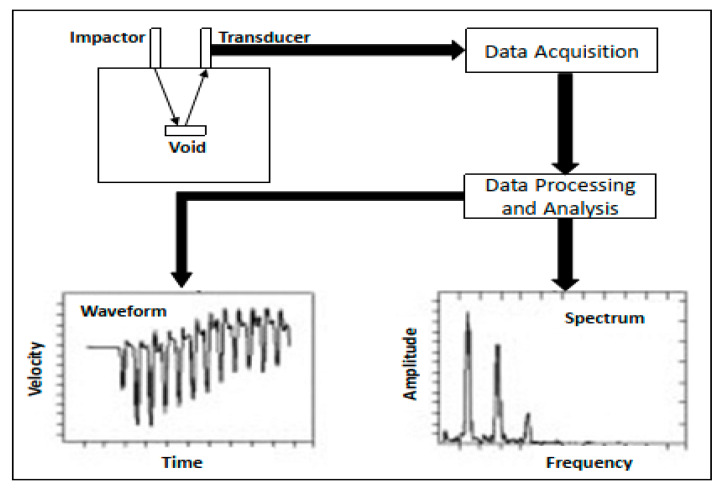
Different elements of the IE method for NDE of infrastructure [111].

**Figure 9 sensors-20-03954-f009:**
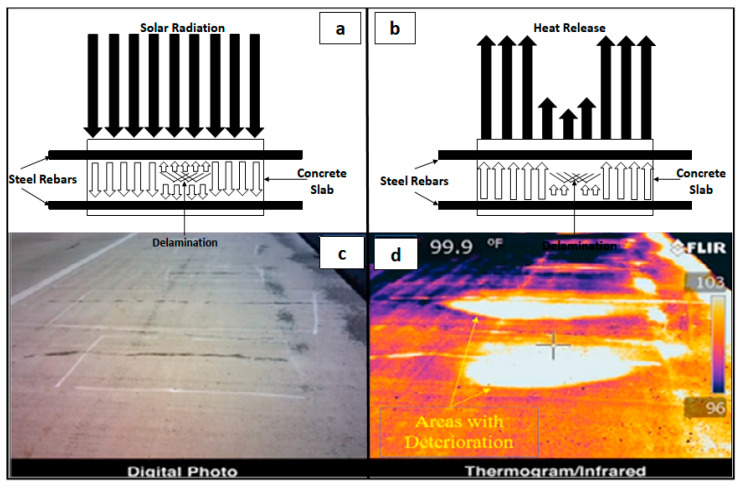
Data collection using a thermography sensor: (**a**) the absorption of solar radiation by the different parts of the infrastructure during the day time [133], (**b**) the emission of radiation from the different parts of the infrastructure during night time [133], (**c**) the output of the data collection unit in the form of a digital image [134], and (**d**) the output of the data collection unit. [134].

**Figure 10 sensors-20-03954-f010:**
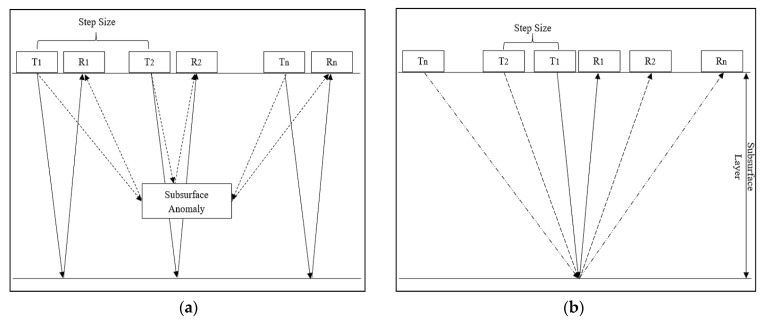
Principles for the GPR wave transmission: (**a**) wave transmission with the fixed-offset profiling method for data collection, (**b**) the use of a common midpoint method for data collection using GPR [149,150] with different visual sensors to provide surface-level information to assess the structural health of bridges [72,74,75,76,77,83,98].

**Table 1 sensors-20-03954-t001:** A complete list of the abbreviations and their meanings that have been used in this review-based study.

Abbreviation	Meaning	Abbreviation	Meaning
LTBP	Long-Term Bridge Performance Program	ROCIM	Robotic Crack Inspection and Mapping
FHWA	Federal High Way Administration	RABIT	Robotics-Assisted Bridge Inspection Tool
NDE	Non-Destructive Evaluation	USV	Unmanned Submersible Vehicle
NDT	Non-Destructive Testing	UGV	Unmanned Ground Vehicle
SHM	Structural Health Monitoring	UMV	Unmanned Marine Vehicle
IR	Infrared	UAV	Unmanned Aerial Vehicle
GPR	Ground-Penetrating Radar	RGB	Red Green Blue
UAS	Unmanned Aerial Systems	RGB-D	Red Green Blue Depth
IE	Impact-Echo	EM	Electro-Magnetic
ER	Electrical Resistivity	USW	Ultrasonic Surface Waves
PRISMA	Preferred Reporting Items for Systematic Reviews and Meta-Analyses	ASTM	American Society of Testing and Materials
HA	Habib Ahmed	CCTV	Closed Circuit Television
HML	Hung Manh La	CNN	Convolutional Neural Network
NG	Nenad Gucunski	RANSAC	Random Sample Consensus
YSI® UUV	YSI® Unmanned Undersea Vehicle	Sea-RAI	Sea Robot Assisted Inspection
BRIDGE	Bridge Risk Investigation Diagnostic Grouped Exploratory	LRF	Laser Range Finder
ETH	Eidgenössische Technische Hochschule	MRC IN-II	Multifunctional Robotic Crawler for Inspection-II
LIDAR	Light and Radar	ABI	Auto Bridge Inspection
ROV	Remotely Operated Vehicle	CCD	Charged Coupled Device
SONAR	Sound Navigation and Ranging	BYU	Bringham Young University
YOLO	You Only Look Once	SVM	Support Vector Machine
HOG	Histogram of Oriented Gradient	FCN	Fully Connected Network
NN	Neural Network	VGG	Visual Geometry Group
ARA	Advanced Robotics and Automation	ResNet	Residual Network

**Table 2 sensors-20-03954-t002:** A number of the review and survey studies that have been published which are relevant to the NDE of civil infrastructure.

Author	Year	Focus	Limitations
Wilson et al. [19]	2007	This review explores different algorithms specific to landmine detection using GPR sensor.	The findings are not up to date. This review focuses on algorithms for data analysis for a single type of NDT/NDE sensor, i.e., GPR. The algorithms highlighted are restricted to a single application, i.e., landmine detection.
Liu et al. [14]	2007	A comprehensive evaluation of different NDT data fusion techniques. The survey also provides brief details regarding the performance evaluation and data visualization for selected applications.	The findings are not up to date. This survey does not include studies related to NDT data fusion techniques for infrastructures. This review does not examine data analyses techniques for single-sensor-based systems.
Khaleghi et al. [15]	2013	The review focuses on examining multi-sensor data fusion methodologies. The review explores different data-related issues (correlated, uncorrelated, inconsistent, conflicting, imperfect data) and algorithms for tackling specific data-related issues.	The findings are not up to date. This study does not attempt to link different issues and their algorithmic solutions with specific practical applications. The review does not explore specific issues related to NDT sensory data and relevant techniques.
Sakagami [13]	2015	This review examines the different NDE techniques for steel bridge cracks using IR thermography. The different surface-level and sub-surface-level algorithms for crack detection using IR thermography have been discussed.	This review only focuses on algorithms for data analysis for a single type of NDT/NDE sensor, i.e., IR thermography. The data analysis algorithms are restricted to a single application, i.e., crack detection in steel bridges.
Rehman et al. [20]	2016	The review examines different NDT methods for bridges using different sensors. For each NDT method, different applications and limitations are discussed.	The review does not assess the state-of-the-art in robotic NDT methods for bridges.
Chen et al. [16]	2018	The review examines construction automation using text mining approach. A clustering-based visualization of relevant research areas has been outlined.	The review does not focus on the various critical qualitative aspects (e.g., types of sensors, platforms, algorithms, and performance evaluation) for construction automation.
Rakha et al. [21]	2018	This review focuses on UAS applications towards SHM for building inspection.	This review only focuses on algorithms for data analysis for a single NDT/NDE sensor, i.e., IR thermography sensors. This review only focuses on the applications based on a single type of robotic platform, i.e., aerial robots.
Feng et al. [22]	2018	This review highlights different vision-based solutions and applications for SHM. The fundamental principles of vision-based systems are discussed (e.g., template matching, coordinate conversion, practical issues related to camera calibration).	This review only focuses on a single type of NDT/NDE sensor, i.e., vision-based sensors. The review is restricted to a single type of SHM application, i.e., visual inspection. The review does not highlight vision-based state-of-the-art systems for SHM.
Agnisarman et al. [17]	2019	This survey examined semi-autonomous systems developed for the visual inspection for the SHM of civil infrastructures. The different studies were classified in terms of application, autonomy, type of visual sensor used, navigational capabilities and algorithms for data analysis.	This review only focuses on algorithms for data analysis for a single type of NDT/NDE sensor, i.e., vision-based sensors. The data analysis algorithms focus on a single type of applications, i.e., visual inspection methods. This review only focuses on algorithms developed for buildings’ energy auditing applications.
Taheri [23]	2019	This review directly explores some of the different sensors used for SHM of concrete structures. For each type of sensor, their various benefits and drawbacks have also been discussed.	The different types of sensors highlighted are restricted to a single type of application, i.e., sensors for measuring and monitoring different properties of concrete. This review does not highlight the different data analysis algorithms for different sensors for SHM.
Martinez et al. [18]	2019	Scientometric review of vision-based systems developed for construction automation has been highlighted. A clustering-based visualization of relevant research areas is provided, along with focus towards keywords, authors, journals, country networks and author networks.	The review does not focus on the qualitative aspects of visual inspection solutions (e.g., type of sensors, types of algorithms, system performance). This review only examines the studies related to a single type of NDT/NDE sensor, i.e., vision-based sensors.
Dabous et al. [24]	2020	This review examines tools and techniques for four non-contact sensors (GPR, IR, laser scanners, photogrammetry) for SHM of bridges. The review also examines some of the different challenges towards the use of four non-contact sensors for the SHM of bridges.	The review only focuses on the platforms and systems developed using a single type of sensory modality, e.g., non-contact-based testing technologies.

**Table 3 sensors-20-03954-t003:** Comparison in the research areas being covered by the different review studies.

Study	Platforms		NDE Sensors					NDE Sensor Fusion		SHM Data Analysis
	Robot	Human	IR	IE	ER	GPR	Vision	General	NDE-Centric	
Wilson [19]										
Liu [14]										
Khaleghi [15]										
Sakagami [13]										
Rehman [20]										
Chen [16]										
Feng [22]										
Rakha [21]										
Agnisarman [17]										
Taheri [23]										
Martinez [18]										
Dabous [24]										
This review										

The green shaded region means that that particular area has been covered by the respective review paper

**Table 4 sensors-20-03954-t004:** Comparison between the state-of-the-art NDE robotic platforms.

Robotic Platforms	Robot Type	NDE Sensors			
		Radar	Vision	Acoustics	Electric
RABIT [12,40,47,49,51,70,79,88,89,99]	Wheeled	1 GPR Array	1 Canon ^®^ Camera	IE and USW Arrays	1 ER Probe
ROCIM [51,88]	Wheeled		1 Canon ^®^ Camera		
ARA Lab Robot [37,50,90,91]	Wheeled	1 GPR Array	1 PrimeSense ^®^ Camera 1 Stereo Camera		2 ER Arrays
ETH Zurich Climbing Robot [59]	Climbing				1 Half-Cell Potential Mapper
BridgeBot [39]	Climbing		1 ArduCAM^®^		
Steel Bridge Climbing Robot [52,100,101]	Climbing		2 Cameras		1 Eddy Current Sensor
ABI Robot [56]	Climbing		1 Camera		
Caterpiller robot [58]	Climbing				
SkySweeper [71]	Climbing				
Cable Robot [57]	Sliding		4 CCD Cameras		
Cable Inspector [64,66]	Climbing		3 Cameras		
CCRobot-II [53,54,63]	Climbing				
MRC^2^ IN-II [68]	Sliding		1 Camera		
Quadrotor platform [81]	UAV				
Manipulator robot [44]	UAV				
Contact prism robot [84]	UAV				
Flying/walking platform [43]	Hybrid				
Octo-rotor platform [45]	UAV				
Quadrotor platform [76]	UAV		1 Camera		
Hammering platform [85]	UAV			1 IE sensor	
3D Mapper Robot [98]	UAV		1 3D LIDAR1 Camera		
DJI® Phantom [74]	UAV		1 Camera		
2D LRF Robot [77]	UAV		1 2D LRF with Mirrors		
Omni-Wheel Robot [83]	UAV		2 Cameras		
Infrared Imagery UAV [72]	UAV		1 IR camera1 RGB camera		
Underwater ROV [80]	USV		1 Camera		
Sea-RAI [41]	USV		4 Cameras		
Muddy Waters [42]	USV		1 Stereo RGB-D Camera1 ARIS® Sonar		
Videoray ROV [41]	USV		1 Camera1 Imaging Sonar		
YSI® Ecomapper [41]	USV		1 Side-Scan Sonar		

Red-shaded region shows that the specific type of NDE sensor was not incorporated in the particular robot platform developed.

**Table 5 sensors-20-03954-t005:** Comparison between the state-of-the-art technique for concrete crack detection.

Study	Application	Dataset	Image Size	Algorithm	Performance
Li et al. [183]	Bridge	1000	N/A	Image Segmentation Algorithm	Accuracy: 92.6% Mean Error: 0.03 mm
Fujita et al. [184]	General	60	640 × 480	Locally Adaptive Thresholding	AuC: 98.0%
Chen et al. [185]	Bridge	40	3088 × 2056	Self-Organizing Map Optimization	Accuracy: 89–91%
Oh et al. [186]	Bridge	80	640 × 480	Morphological Operations	Accuracy: 94.1%
Li et al. [187]	Bridge	1200	4288 × 2848	Active Contour Model with SVM	Width Accuracy: 92.1% Mean Error: 0.03 mm
Liu et al. [188]	General	84	512 × 512	U-Net	Precision: 90.0% Recall: 91.0% F1-score: 90.0%
Ren et al. [152]	Tunnels	409	4032 × 3016	CrackSegNet	Precision: 66.07% Recall: 85.54% F1-score: 74.55%
Dung et al. [189]	General	40,600	227 × 227	FCN with VGG-16	Precision: 90% Max-F1: 90%
Zhou et al. [157]	Road	52,408	256 × 256	ResNet	Precision: 99.7% Recall: 99.8% F1-score: 99.8%
Billah et al. [180]	Bridge	43,996	256 × 256	ResNet-50	Accuracy = 94.0%
Park et al. [164]	General	1800	N/A	YOLO-V3-tiny	Thickness Error: 0.09 mmLength Error: 2.72 mm
Billah et al. [182]	Bridge	12,000	256 × 256	SegNet	Accuracy = 98.7% Error = 1.3% F1-score = 24.1%
Li et al. [151]	Sewer	18,333	224 × 224	ResNet-18 with Hierarchical SoftMax	Accuracy: 64.8%
Wang et al. [154]	Ceiling	1953	400 × 600	DCNN	Accuracy: 86.22%

**Table 6 sensors-20-03954-t006:** Comparison between the state-of-the-art techniques for rebar detection and localization.

Study	Features	Dataset	Rebar Detection Techniques	Performance	Rebar Localization Techniques	Performance
Dou et al. [138]	Edge Features	Synthetic + Real Data	C3 Algorithm + 3-Layer Feed-Forward NN	Recall: 0.704 Precision: 0.708	Orthogonal Hyperbola Fitting	Time: 0.73 s/rebar
Kaur et al. [47]	HOG	3 Bridges	SVM	Acc.: 91.98%	RANSAC + Hyperbola Fitting	Accuracy: 91.98%
Gibb et al. [135]	HOG	4 Bridges	Naïve Bayes	Acc.: 95.05%	Precise Hyperbola Localization	Time: 32.4 s/image
Dinh et al. [136]	N/A	26 Bridges	24-layer CNN	Acc.: 99.6%	Rebar Picking	Accuracy: 99.6%
Ahmed et al. [138]	N/A	6 Bridges	ResNet-50	Acc.: 99.42%	K-Means Clustering	Accuracy: 94.52% Precision: 95.18%
Harkat et al. [240]	HOS cumulant	133 radargrams	MOGA + 3-layer CNN	Acc.: 88.99%	Hough Transform	N/A
Ahmed et al. [239]	N/A	9 Bridges	Deep ResNets	Train Acc.: 99.4% Val. Acc.: 97.20%	Novel Rebar Localization Algorithm	Accuracy: 91.91% Precision: 96.89% Recall: 94.41% F1-score: 95.58%

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
