# Peer review of "Review of Non-Destructive Civil Infrastructure Evaluation for Bridges: State-of-the-Art Robotic Platforms, Sensors and Algorithms"

_sensors, 2020, doi:10.3390/s20143954_

Round 1
Reviewer 1 Report
This paper provides a comprehensive survey of the state-of-the-art of non-destructive civil infrastructural evaluation for bridges, including the summary of robotic platforms, sensors, and related algorithms. It is a very comprehensive and detailed review paper. It is a topic of interest to researchers in the related areas and will help new researchers identify starting topics for non-destructive evaluation. But the main problem of this paper is that too board coverage leads to a lack of emphasis. It will be difficult to read. I will suggest authors delete or simplify some sections, and then add a figure after line 70 to show the clear logic relationship of your 9 sections. Another problem is the limitation of other studies is not summarized properly in table 1.
Meanwhile, there were significant grammatical and organizational errors throughout the paper. These must be corrected and brought up to an acceptable standard. Some specific examples are included below. This is not an exhaustive list but is provided as examples.
- Line 9: there is a font error.
- Line 23-25: too many keywords; Punctuation between keywords should be the same.
- Line 112: The full name needs to be given when the abbreviation first appears in the text, such as IE, IR, GPR, ER.
- Line 120, the authors claim green shaded region but it is grey in the table
- Figure 2 is not very clear, and the text in the figure is elongated.
- Paragraph indentation error should be revised throughout the paper, such as line 215-216, 556-559, 168-177 etc.
- Line 218 and 223: the citation should be [31-35] and [38-47], respectively. Similar mistakes should be revised throughout the paper.
- Line 227-231: line spacing error.
Author Response
Reviewer Comment 1: This paper provides a comprehensive survey of the state-of-the-art of non-destructive civil infrastructural evaluation for bridges, including the summary of robotic platforms, sensors, and related algorithms. It is a very comprehensive and detailed review paper. It is a topic of interest to researchers in the related areas and will help new researchers identify starting topics for non-destructive evaluation. But the main problem of this paper is that too board coverage leads to a lack of emphasis. It will be difficult to read. I will suggest authors delete or simplify some sections, and then add a figure after line 70 to show the clear logic relationship of your 9 sections.
Authors’ Response 1: The authors are of the opinion that all of the sections included in the discussion are deeply inter-twined and part of the novelty that is being highlighted in this research. Therefore, the focus is towards enhancing clarity of the discussion.
In accordance with the reviewer comment, a new figure has been developed (figure 2), which provide a roadmap from section 1 to section 9, which can enable the readers to better understand the relationship between the different sections (Before Line 52, Page 3)
Reviewer Comment 2: Another problem is the limitation of other studies is not summarized properly in table 1
Authors’ Response 2: Table 2 has been modified with all of its internal content (After line 113, page 4)
Meanwhile, there were significant grammatical and organizational errors throughout the paper. These must be corrected and brought up to an acceptable standard. Some specific examples are included below. This is not an exhaustive list but is provided as examples.
Reviewer Comment 3: Line 9: there is a font error.
Authors’ Response 3: The error has been removed in accordance with the reviewer’s suggestion (Line 12, Page 1)
Reviewer Comment 4: Line 23-25: too many keywords; Punctuation between keywords should be the same
Authors’ Response 4: According to the Sensors template, ten keywords are allowed. The authors have put ten keywords to ensure that all of the major aspects of the research can be highlighted. All of the keywords are separated by semi-colon punctuation (Line 29-32, Page 1).
Reviewer Comment 5: Line 112: The full name needs to be given when the abbreviation first appears in the text, such as IE, IR, GPR, ER.
Authors’ Response 5: The requested change has been made in the manuscript (Line 147-148, Page 7). A complete list of keywords used in the review paper has also been provided in the first section of the manuscript (After Line 51, Page 2).
Reviewer Comment 6: Line 120, the authors claim green shaded region but it is grey in the table
Authors’ Response 6: The shading in the table has been changed to match the table footnote (After Line 122, Page 6).
Reviewer Comment 7: Figure 2 is not very clear, and the text in the figure is elongated
Authors’ Response 7: The figure dimensions have been changed so that the internal text is clear to the readers, in accordance with the suggestion of the reviewer (Before Line 173, Page 8)
Reviewer Comment 8: Paragraph indentation error should be revised throughout the paper, such as line 215-216, 556-559, 168-177 etc.
Authors’ Response 8: Indentation errors all across the manuscript have been eradicated.
Reviewer Comment 9: Line 218 and 223: the citation should be [31-35] and [38-47], respectively. Similar mistakes should be revised throughout the paper.
Authors’ Response 9: The issues with the referencing have been incorporated throughout the manuscript
Reviewer Comment 10: Line 227-231: line spacing error.
Author’s Response 10: The line spacing error has been removed all across the manuscript
Reviewer 2 Report
This article is meta-review, that is, a review of reviews, on non-destructive civil infrastructure evaluation for bridges. The topic is as timely as ever, with the decay in infrastructure underlined by everyday experience and professional assessment alike.
Unfortunately, the authors fail to convince me for the need of this present study. This is partly due to somewhat hurried presentation. There are many mistakes both in language and organisation of the paper which gives an impression of sufficient lack of detail in internal review or work under some deadline. For instance, the authors use abbreviations without definitions or only define them in figure captions. For readers of Sensors this is perhaps acceptable but one should follow best practices even when writing for an expert audience.
My suggestion is that the authors, who clearly know their field well, spend some time on defining the message of this work. This message should be reflected in the introduction where the research highlights should be clearly stated. In fact, after careful reading I still cannot state precisely what the scientific contribution of this paper is. Even the conclusions fail to address this question.
I want to end on a positive note, however. There is a good paper buried somewhere beneath the version presented here. With a little bit of extra work and definition of the goals this study will find its intended audience.
Author Response
Reviewer Comment 1: This article is meta-review, that is, a review of reviews, on non-destructive civil infrastructure evaluation for bridges. The topic is as timely as ever, with the decay in infrastructure underlined by everyday experience and professional assessment alike.
Authors’ Response 1: The authors are of the opinion that the manuscript has been developed as a review paper, not a meta-review. Majority of the papers included in the paper are original researches that have been developed in the recent past in the field of non-destructive evaluation of civil infrastructure in general and bridges in particular. Section 2 provides a comparison between the different relevant review papers, so that the novelty of the present review can be better appreciated.
Reviewer Comment 2: Unfortunately, the authors fail to convince me for the need of this present study. This is partly due to somewhat hurried presentation. There are many mistakes both in language and organization of the paper which gives an impression of sufficient lack of detail in internal review or work under some deadline. For instance, the authors use abbreviations without definitions or only define them in figure captions. For readers of Sensors this is perhaps acceptable but one should follow best practices even when writing for an expert audience.
Authors’ Response 2: The errors in language and organization of the paper have been eradicated. With regards to abbreviation, table 1 has been added, which provides a comprehensive list of different abbreviation and their meanings (After Line 51, Page 2). Also, the first mention of all abbreviations have been provided in their complete form throughout the manuscript.
Reviewer Comment 3: My suggestion is that the authors, who clearly know their field well, spend some time on defining the message of this work. This message should be reflected in the introduction where the research highlights should be clearly stated. In fact, after careful reading I still cannot state precisely what the scientific contribution of this paper is. Even the conclusions fail to address this question.
Authors’ Response 3: In accordance with the reviewer suggestion, the motivation of the manuscript has been clearly articulated in the introduction section (Line 70-84, Page 3-4). At the same time, the novelty aspect and contributions have been further enhanced in the conclusion section (Line 770-776, Page 25).
I want to end on a positive note, however. There is a good paper buried somewhere beneath the version presented here. With a little bit of extra work and definition of the goals this study will find its intended audience.

Reviewer 3 Report
This review-based study presented the state-of-the-art platforms that have been developed for bridge inspection and evaluation. Moreover, the review methodology has also been discussed in sufficient depth. This paper is mostly well written and the overall organization is clear, as well as the main results of the research. There are some format errors scattered in the content that should be carefully checked and corrected. Some of them are listed here:
- At line 67, ‘Section 7 will highlight the’ is not a completed sentence.
- The sentences between line 121 and line 132 repeats the previous paragraph.
- At line 212, the title should be section 4 instead of 1. The same error occurs at line 244, which should be section 5 instead 1. It also happens at line 699, and it should be section 9.
The above points should be modified by the authors. Overall, this paper is accepted with minor revision.
Author Response
This review-based study presented the state-of-the-art platforms that have been developed for bridge inspection and evaluation. Moreover, the review methodology has also been discussed in sufficient depth. This paper is mostly well written and the overall organization is clear, as well as the main results of the research. There are some format errors scattered in the content that should be carefully checked and corrected. Some of them are listed here:
Reviewer Comment 1: At line 67, ‘Section 7 will highlight the’ is not a completed sentence.
Authors’ Response 1: The sentence has been completed (Line 94-95, Page 4).
Reviewer Comment 2: The sentences between line 121 and line 132 repeats the previous paragraph
Authors’ Response 2: The repetitive sentences have been removed (Line 139-153, Page 7)
Reviewer Comment 3: At line 212, the title should be section 4 instead of 1. The same error occurs at line 244, which should be section 5 instead 1. It also happens at line 699, and it should be section 9.
Authors’ Response 3: The section numbering has been changed (Line 245, Page 10)
The above points should be modified by the authors. Overall, this paper is accepted with minor revision.

Round 2
Reviewer 1 Report
The authors have replied all my questions. This paper can be accepted in the current version.